# Protocol for a prospective observational cohort study to assess clinical applications of expanded noninvasive prenatal testing (NIPT) in pregnancies with placental dysfunction

Ana Villalba[1], M. Ángeles Sánchez-Durán[2,3], Carmen Orellana[4], Mercedes Sobreviela[5,6], María M. Gil[7,8], Rosa María Lobo[9], Irene Gómez-Manjón[10,11], Cristina Cea[12,13], Laia Pedrola Vidal[4], Sebastian Menao[14], Raquel Martin-Alonso[7,8], Gonzalo E. Quesada[15], Eva Albuixech[12], Mónica Roselló Piera[4], Eva Valle[5], Eduardo Tizzano[12,13], Alberto Galindo[1,11,16,17], Manel Mendoza[2,3], Beatriz Marcos[4], José Vicente Cervera[4], Belén Santacruz[7,8], Daniela Piazzolla[18], Lieve Page-Christiaens[18,19], Diego Lerma[5], Ramiro Quiroga[20], Francisco Javier Fernández[10,11], Ignacio Herraiz[1,11,16,17] *

1 Department of Obstetrics and Gynecology, Hospital Universitario 12 de Octubre, Madrid, Spain, 2 Department of Obstetrics, Vall d'Hebron Barcelona Hospital Campus, Barcelona, Spain, 3 Departament de Pediatria, Obstetrícia i Ginecologia i de Medicina Preventiva i Salut Pública, Universitat Autònoma de Barcelona, Bellaterra, Spain, 4 Genetics Unit, Hospital Universitari i Politècnic La Fe, Valencia, Spain, 5 Department of Obstetrics, Hospital Clínico Universitario Lozano Blesa, Zaragoza, Spain, 6 Aragon Institute of Health Research (IIS Aragon), Zaragoza, Spain, 7 Department of Obstetrics and Gynecology, Hospital Universitario de Torrejón, Torrejón de Ardoz, Madrid, Spain, 8 School of Medicine, Universidad Francisco de Vitoria, Madrid, Spain, 9 Clinical Analysis Service, Hospital Río Hortega, Valladolid, Spain, 10 Department of Genetics, Hospital Universitario 12 de Octubre, Madrid, Spain, 11 Instituto de Investigación Hospital 12 de Octubre (imas12), Madrid, Spain, 12 Department of Clinical and Molecular Genetics, Hospital Universitari Vall d'Hebron, Barcelona, Spain, 13 Medicine Genetics Group, Vall d'Hebron Research Institute (VHIR), Barcelona, Spain, 14 Biochemistry Department, Hospital Clínico Universitario Lozano Blesa, Zaragoza, Spain, 15 Department of Obstetrics and Gynecology, Hospital Río Hortega, Valladolid, Spain, 16 Facultad de Medicina, Universidad Complutense de Madrid (UCM), Madrid, Spain, 17 Primary Care Interventions to Prevent Maternal and Child Chronic Diseases of Perinatal and Developmental Origin (RICORS), RD21/0012/0024, Instituto de Salud Carlos III, Madrid, Spain, 18 Medical Affairs Europe, Illumina, Inc., Great Abington, Cambridge, United Kingdom, 19 Department of Obstetrics and Gynecology, University Medical Center Utrecht, Utrecht, The Netherlands, 20 Department of Obstetrics and Gynecology, Hospital Universitari i Politècnic La Fe, Valencia, Spain

* ignacio.herraiz@salud.madrid.org

## Abstract

### Introduction

Expanded non-invasive prenatal testing (NIPT) utilizing cell-free DNA (cfDNA) in maternal blood enables the detection of common trisomies as well as rare autosomal trisomies and large deletions and duplications within the genome of the placenta and, therefore, the fetus. Given that cfDNA predominantly originates from trophoblastic cells, expanded NIPT also holds the potential to reveal confined placental mosaicism (CPM). CPM has been associated with placental dysfunction, which can result in reduced levels of the circulating placental biomarkers routinely tested as

**Data availability statement:** Data cannot be shared publicly because they contain sensitive information from human participants, including potentially identifiable clinical and genetic data. Furthermore, the informed consent obtained from participants did not include provisions for public data sharing, and applicable data protection regulations restrict open access to such data. Data are available upon reasonable request for qualified researchers who meet the criteria for access to confidential data, subject to prior review and approval by the Institutional Research Ethics Committee (CEIm) of Hospital Universitario 12 de Octubre. Requests for data access may be directed to the CEIm at: ceic@h12o.es (or to the corresponding institutional contact), which acts as an independent body overseeing data access in compliance with ethical and legal requirements.

**Funding:** This study was funded by Spanish network RICORS-SAMID (RD21/0012/0024): Primary care interventions to prevent maternal and child chronic diseases of perinatal and developmental origin, Instituto de Salud Carlos III, Madrid, Spain, and financed by the European Union though the Next Generation EU funds, which finance the actions of the Recovery and Resilience Facility (RRF). These funders had no role in study design, data collection and analysis, or decision to publish. This work is also supported with reagents, technical, and editorial support from Illumina, Inc. through collaboration agreements with the participating centers. Illumina, Inc. will not contribute to data collection. Contributions to the study design were provided solely by employees of the company who are named as coauthors.

**Competing interests:** The authors report receiving reagents, technical, and editorial support, through collaboration agreements between the participating centers and Illumina, Inc. At the time when this protocol was under development, Lieve Page-Christiaens was an employee and shareholder of Illumina, Inc. Daniela Piazzolla is an employee and shareholder of Illumina Inc. This does not alter our adherence to PLOS ONE policies on sharing data and materials.

part of first-trimester aneuploidy screening and in early-onset fetal growth restriction (eoFGR). This study aims to explore whether expanded NIPT adds clinical value compared with targeted NIPT in pregnancies exhibiting placental dysfunction, which is proxied herein by first-trimester pregnancy-associated plasma protein A (PAPP-A) or the beta-subunit of human chorionic gonadotropin (β-hCG) values <0.3 multiples of the median (MoM) or the diagnosis of eoFGR.

## Materials and methods

This is a prospective, multicenter, observational cohort study conducted at six Spanish sites from 2021 to 2026. Inclusion criteria are singleton pregnancies with PAPP-A or β-hCG values <0.3 MoM in the first-trimester or eoFGR defined as FGR diagnosed before 32 + 0 weeks of gestation in the absence of congenital anomalies according to Delphi criteria. Expanded NIPT is performed in all enrolled patients, and pregnancy and perinatal outcomes are evaluated.

## Protocol version

6.0, 27th January 2026

## Introduction

Noninvasive prenatal testing (NIPT) was first introduced into clinical practice in 2011 and has had a major impact on prenatal diagnosis ever since. Targeted NIPT assays cell-free DNA (cfDNA) in the maternal blood, providing the most sensitive and specific screening available for detecting common fetal trisomies such as trisomy 21, trisomy 18, and trisomy 13, and sex chromosome abnormalities [1,2]. Compared to the pre-NIPT era, the utilization of NIPT has contributed to a 38% reduction in invasive procedures performed on pregnant women for whom biomarker screening suggested a high risk of a fetus with Down syndrome [3].

Over time, advancements in technologies have expanded the capabilities of NIPT, enabling the detection of rare autosomal trisomies (RATs) and large deletions and duplications, known also as copy-number variations (CNVs). Accumulating data demonstrates the utility of expanded NIPT for detecting a broad range of clinically significant genetic diseases [4,5]. However, expanded NIPT's relatively low positive predictive value (PPV) for RATs and CNVs—approximately 5% [6] and 11%−50% [7,8], respectively, with high variability depending on the patient's *a priori* risk—along with the greater complexity of its analysis, interpretation, and counseling, has hindered its widespread adoption in clinical practice. This contrasts with women´s preferences, who choose expanded over targeted NIPT in 84% of cases, according to a recent study [9].

NIPT is based on detecting cfDNA originating from the placenta in maternal blood. As a result, complete or mosaic RATs or large CNVs detected by NIPT may reveal fetal chromosomal abnormalities associated with abnormal fetal phenotypes as well as confined placental mosaicism (CPM). CPM is a condition in which, based on

cytogenetic studies of chorionic villus sampling (CVS) specimens, the placenta and fetus have different chromosomal constitutions, usually with a chromosomally normal embryo and placental trisomy or, less frequently, placental CNVs. This condition represents the main cause of false positive results of expanded NIPT [10]. In contrast, true fetal mosaicism (TFM) is a condition where the placenta and the fetus have two or more populations of cells with different genetic make-ups. Expanded NIPT is more sensitive than CVS for detection of low-level CPM involving the cytotrophoblast [11], which is associated with adverse pregnancy outcomes related to placental dysfunction including fetal growth restriction (FGR) [12]. Previous studies have shown that, in pregnancies with FGR, the rate of CPM is approximately twice that of normally growing fetuses and the earlier the onset of FGR, the higher the rate of CPM [13,14].

In the first trimester of pregnancy, even before the onset of FGR, certain detectable biochemical markers act as predictive indicators of high risk of placental dysfunction. These markers include two placental hormones, pregnancy-associated plasma protein A (PAPP-A) and the beta-subunit of human chorionic gonadotropin (β-hCG), which are routinely tested as part of first-trimester screening for fetal aneuploidy. When the values of these hormones are low—below 0.3 multiples of the median (MoM), which is equivalent to the third percentile—the risk of FGR is four times higher compared to the general population [15]. Furthermore, a relationship between CPM and low PAPP-A has been demonstrated [16]. The rate of positive screening for RATs and large CNVs in a population at high risk of chromosomal diseases, based on the results of the first-trimester combined test, is around 1.6% compared to less than 0.5% in the low-risk population [17]. Therefore, pregnancies with low PAPP-A could be a target population for the detection of FGR-related CPM.

Once the diagnosis of FGR has been established, the ability to identify or exclude an underlying genetic cause may inform prenatal as well as postnatal follow-up. In cases of early-onset fetal growth restriction (eoFGR), defined as occurring before 32 weeks of gestation, amniocentesis is often recommended for genetic diagnosis, despite placental dysfunction being a likely cause. In such cases, NIPT may offer additional information about CPM that cannot be obtained through the analysis of fetal cells in amniotic fluid [18]. Additionally, NIPT results may be informative if amniocentesis is not technically feasible, for instance in case of oligohydramnios, or if the woman declines invasive diagnostic testing.

The ratio between the angiogenic biomarkers soluble fms-like tyrosine kinase-1 (sFlt-1) and placental growth factor (PlGF) has also been shown to correlate with the degree of placental dysfunction and severity of eoFGR [19]. A better understanding of the relationship between expanded NIPT results and the sFlt-1:PlGF ratio could lend support for the clinical value of adding expanded NIPT to the workup of eoFGR.

This study investigates the role of expanded NIPT in detecting fetal or placental chromosomal abnormalities when placental dysfunction is suspected due to abnormal biochemical markers in the first trimester of pregnancy or isolated eoFGR in the second/third trimesters.

## Materials and methods

### Study setting

This is a prospective, multicenter, observational cohort study that is conducted at six Spanish sites including Hospital Universitario 12 de Octubre and Hospital de Torrejón in Madrid, Hospital Universitario Vall d'Hebron in Barcelona, Hospital Río Ortega in Valladolid, Hospital Universitari i Politècnic La Fe in Valencia, and Hospital Lozano Blesa in Zaragoza.

### Dates of the study

The study is planned to run from 2021 to 2026 and is currently in the recruitment phase. Participant recruitment began on October 5, 2021, and is expected to be completed by mid-2026. Data collection will continue in parallel and is planned to be finalized by September 2026. The analysis of the collected data is scheduled to be performed thereafter, with results anticipated by December 31, 2026.

## Objectives

**Study aim.** To determine whether expanded NIPT adds clinical value compared with targeted NIPT in cases of placental dysfunction detected on the basis of first-trimester PAPP-A and/or β-hCG values <0.3 MoM or early diagnosis of FGR.

**Study objectives.** The primary objective of this study is to determine the prevalence of genetic alterations identified by expanded NIPT, including TFM and CPM, in pregnancies with PAPP-A and/or β-hCG < 0.3 MoM or isolated eoFGR, and subsequently to compare that prevalence with the prevalence of genetic alterations detected by targeted NIPT.

Secondary objectives of this study include:

To assess the rates of fetal congenital anomalies (as assessed either prenatally or postnatally up to the age of six months) associated with:

- High-risk expanded NIPT results for RATs and CNVs

- High-risk NIPT results for common aneuploidies

- Low-risk NIPT results

To evaluate the percentage of eoFGR in women with first trimester PAPP-A < 0.3 MoM and/or β-hCG < 0.3 MoM with:

- High-risk expanded NIPT results for RATs and CNVs

- High-risk NIPT results for common aneuploidies

- Low-risk NIPT results

To investigate the clinical course and perinatal outcomes of pregnancies included in this study

## Inclusion criteria

Eligible patients are women aged 18 years or older with ongoing singleton pregnancies who meet at least one of the following criteria: PAPP-A < 0.3 MoM in the first-trimester screening for aneuploidy; β-hCG < 0.3 MoM in the first-trimester screening for aneuploidy; or eoFGR in the absence of evidence of congenital anomalies. Patients will only be included if they either wish to receive the results of expanded NIPT prenatally or, if they do not wish to receive the results of expanded NIPT (pre- or postnatally), consent to having their plasma samples frozen and processed after delivery, provided the enrolling center has appropriate facilities for frozen storage.

## Exclusion criteria

Patients who do not provide written informed consent to participate in the study are excluded. For the FGR group, patients are also excluded if there is prenatal evidence of a major congenital anomaly or a prenatal congenital infection.

## Definition of eoFGR

EoFGR is defined as FGR diagnosed before 32 + 0 weeks of gestation in the absence of evidence of congenital anomalies and meeting any of the following three criteria [20]:

- Abdominal circumference or estimated fetal weight <3rd percentile

- Absence of end-diastolic flow in the umbilical artery

- Abdominal circumference or estimated fetal weight <10th percentile with uterine artery pulsatility index >95th percentile and/or umbilical artery pulsatility index >95th percentile

## Study regimen

The study will include the following steps:

- Identify pregnant women who meet the eligibility criteria.

- Inform the participants about the study, invite them to participate, and explain the different options regarding the type of information they can choose to receive (including information about findings that might be difficult to interpret such as CNVs of uncertain significance) and the timing (as soon as it is available or after delivery).

- Ascertain expert genetic counseling and support in case of difficult genetic results.

- Record the number and data of eligible volunteers who choose not to participate, but excluding identification data.

- Obtain written informed consent, after which recruitment will become effective. The consent form is signed in the presence of an investigator, who provides a verbal explanation of the purpose of the study, risks, and benefits in plain language, and once the participant voluntarily agrees to participate. The signed consent is stored in paper format within the departments for potential future verification.

- Pseudo-anonymization: assign a unique study number to the patient's electronic study data record.

- Draw 10 ml of blood and transport it to the laboratory for NIPT. Blood draws are to be performed in the two weeks following the diagnosis of eoFGR or detection of PAPP-A and/or β-hCG < 0.3 MoM but not before 10 weeks gestational age.

- Perform NIPT and report the NIPT results in accordance with the patient's choice regarding the receipt of results and the timing (prenatally or postnatally). If the initial NIPT analysis yields no result, the reason for the test failure will be assessed and a fresh blood sample will be requested. If a repeat test fails again from the fresh sample, the participant will not be excluded from the study but instead the test outcome will be classified as invalid and counted among the study results as a final no-call result.

- Monitor the pregnancy and eventually conduct invasive diagnostic testing as per routine clinical practice.

- If there are discrepancies between the genetic results of NIPT and invasive test results, collect two to four biopsy specimens from different quadrants of the placenta on the fetal side after delivery; in the event of pregnancy termination or fetal demise, collect the products of conception (fetal and placental), whenever possible, for the purpose of genotyping.

- In the event of abnormal NIPT with no invasive prenatal test results, collect two to four biopsy specimens from different quadrants of the placenta on the fetal side after delivery; these specimens will be used for genotyping.

- Conduct physical examination of the newborn by an experienced neonatologist, or by a pathologist in case of perinatal death; if an unusual phenotype is observed, and if parental consent is provided, perform genotyping of the newborn according to routine clinical practice.

- Monitor the newborn's development for a period of at least six months to rule out evidence of genetic disease during this period.

- Record all study variables, including pregnancy and delivery outcomes, postnatal follow-up, interventions or abstention of interventions, and deviations from standard pregnancy care in the electronic clinical research database (CRD) created for this study (S1 Appendix: Description of study variables and data collection).

## Analytical methods

**Biochemical determination of PAPP-A and β-hCG in the first trimester.** Maternal blood samples for the determination of PAPP-A and β-hCG in the context of routine first-trimester screening for aneuploidy are collected

between 8 + 0 and 13 + 6 weeks of gestation in tubes with the appropriate anticoagulants as required by the test provider and centrifuged at 2000 g for 10 min within four hours of collection. Samples are analyzed using the following platforms: Brahms Kryptor® (Thermo Fisher Scientific, Clinical Diagnostics, Brahms GmbH, Henningsdorf, Germany), Hospital 12 de Octubre and Hospital de Torrejón; Roche Cobas or Elecsys® (Roche Diagnostics Deutschland GmbH, Mannheim, Germany), Hospital Vall d'Hebron; Siemens IMMULITE® (Siemens Healthcare GmbH, Erlangen, Germany), Hospital La Fe and Hospital Río Ortega; and DELFIA® Xpress (PerkinElmer, Turku, Finland), Hospital Lozano Blesa. PAPP-A and β-hCG values are converted to MoM using platform-specific software programs.

**Optional: Biochemical determination of sFlt-1 and PlGF at diagnosis of FGR (only at sites where this is done routinely).** sFlt-1 and PlGF are tested on a peripheral blood sample from the pregnant woman drawn at the time of diagnosis of eoFGR (plus or minus three days) and measured using a sandwich immunoassay with two monoclonal antibodies specifically directed against PlGF (Elecsys PlGF, human PlGF) and sFlt-1 (Elecsys sFlt-1, human sFlt-1) using the COBAS or ELECSYS® automated platform (Roche Diagnostics Deutschland GmbH, Mannheim, Germany). Detection limits of the assay are 3 pg/ml for PlGF and 10 pg/ml for sFlt-1.

**Cytogenetic/cytogenomic analysis of CVS samples.** From the two to four placental biopsies, DNA is isolated from the whole villi separately for each villus. The following analyses are performed:

- Quantitative fluorescence polymerase chain reaction (QF-PCR) for the detection of common aneuploidies in each biopsy specimen; maternal contamination is ruled out by comparing the genotype of each sample with the maternal genotype

- Karyotype alone or karyotype plus DNA microarrays using comparative genomic hybridization (CGH) arrays (Agilent 60K and 180K) or single-nucleotide polymorphism (SNP) arrays depending on local availability and the genetic abnormality detected

  ◦ Two long-term cultures are set up for each biopsy specimen to rule out mosaicism due to the culture process

  ◦ DNA for microarrays is extracted directly from the samples

  ◦ If using DNA obtained from the cultured sample, maternal origin is ruled out if the embryo is female

**NIPT using whole-genome sequencing.** VeriSeq™ NIPT Solution v2 (Illumina, Inc., San Diego, CA, USA) is used for non-invasive prenatal testing. This whole-genome sequencing-based NIPT test offers two screening options: the basic (targeted) mode, which screens for chromosomes 13, 18, 21 and sex chromosomes only, and the genome-wide (expanded) mode, which includes targeted NIPT plus screening for RATs and CNVs ≥ 7 Mb in all autosomes. The following quality control metrics will be also reported: fetal fraction, mosaic ratio, aneuploidy log-likelihood ratio score, and mosaic/default log-likelihood ratio score.

**Handling of biological samples.** This study complies with the provisions of the Biomedical Research Law 14/2007 and those established in Order SSI/2065/2014, of October 31st. Analyses are carried out in laboratories at each center except for Hospital Torrejón, which will send study samples to Hospital 12 de Octubre. Biological samples collected for this study will not be stored once their analysis has been completed.

## Statistical analysis

**Sample size estimation.** For pregnancies with eoFGR, our sample size estimations are based on statistical methods for sample size calculation (α error of 0.05 and a power of 80%) and an earlier review by Meler, Sisterna and Borrell [14]. In this review, the risk of chromosomal abnormalities in eoFGR that are detectable with conventional methods was estimated to be 6.4%. It was further assumed that CPM might be present in 9% of cases of eoFGR. Assuming that NIPT can detect 70% of these CPMs, it is estimated that at least 181 subjects are required to demonstrate additional value of

using expanded NIPT to detect genetic abnormalities in the fetus or placenta. Based on a previous survey conducted at participating sites on the number of cases of eoFGR managed every year, it is estimated that this sample size can be reached during the study period.

For pregnancies with PAPP-A < 0.3 MoM and/or β-hCG < 0.3 MoM, based on previous data from CVS, the prevalence of chromosomal mosaicism is between 1.2%–2% in high-risk pregnancies [12,21]. Therefore, we assumed an expected increase of at least 1.0% in the detection of these anomalies with the use of expanded NIPT. To demonstrate these differences, the corresponding necessary sample size is 321 subjects according to the same power analysis.

Given the limited evidence on CPM incidence and false-negative NIPT results, we have increased the calculated sample size for both the eoFGR and PAPP-A/β-hCG groups by 5%. Thus, the final target sample size will be 191 for the eoFGR group and 338 for pregnancies with PAPP-A < 0.3 MoM and/or β-hCG < 0.3 MoM.

**Data collection.** Data collection is performed using a database created for this purpose on the REDCap™ platform [22], available via the Research Institute of Hospital 12 de Octubre (i + 12). The online REDCap™ platform allows case report forms to be created and filled in by authorized sites in real time. The system allows data to be imported or exported in other formats such as Microsoft Excel, PDF, SAS, Stata, R, or SPSS. Data is encrypted and complies with the Personal Data Protection and Guarantee of Digital Rights and Regulation (EU) 2016/679 of the European Parliament and of the Council of April 27, 2016 on Data Protection (GDPR).

**Data analysis.** A descriptive statistical analysis is performed on the demographic, clinical, and laboratory variables of patients. Depending on whether expanded NIPT identifies a genetic abnormality, statistical comparisons are performed using chi-square or Fisher's exact test for qualitative variables and Student's t-test or Mann-Whitney U-test for quantitative variables. P values are estimated using two-tailed tests, with the level of statistical significance set at $p < 0.05$. Measures of association are expressed as relative risk (RR) with a 95% confidence interval (CI). Data analysis is carried out using SPSS Version 20.0 (SSPS Inc., Chicago, IL, USA).

## Ethics

This study protocol was approved by the Ethics Committee of the Hospital Universitario 12 de Octubre in Madrid (Nº CEIm 21/223) on 25 May 2021. Recruited patients and their partners, if applicable, receive both verbal and written information explaining the purpose of the study and are required to provide signed informed consent.

## Discussion

This is to our knowledge the first study designed to evaluate the potential role of expanded NIPT in identifying chromosomal abnormalities, including CPM, that are related to placental dysfunction as suspected by low PAPP-A or β-hCG in the first trimester or the development of eoFGR before 32 weeks of gestation. Currently, there is no clear consensus on how to counsel parents about potential adverse perinatal outcomes associated with the detection of CPM. The findings of this study may contribute to a more accurate interpretation of expanded NIPT results in scenarios suggestive of placental dysfunction—particularly given the growing use of NIPT and its potential to identify CPM linked to such dysfunction.

Due to the rarity of the events under study, the main challenge for this study lies in obtaining a sufficiently large cohort to achieve statistical power. To achieve the necessary sample size, a multicenter strategy involving six participating sites was adopted. An extended recruitment period is also required, which places significant demands on the research team, but at the current pace, it should be possible to obtain results by the end of 2026.

## Supporting information

**S1 Appendix. Description of study variables and data collection** [23,24].
(DOCX)

## Acknowledgments

The authors thank Kristine Jinnett and Raye Alford (Illumina, Inc.) for their editorial support.

## Author contributions

**Conceptualization:** Ana Villalba, M. Ángeles Sánchez-Durán, Carmen Orellana, Mercedes Sobreviela, María M. Gil, Rosa María Lobo, Irene Gómez-Manjón, Cristina Cea, Laia Pedrola Vidal, Sebastian Menao, Gonzalo E. Quesada, Eva Albuixech, Mónica Roselló Piera, Eva Valle, Eduardo Tizzano, Alberto Galindo, Manel Mendoza, Beatriz Marcos, José Vicente Cervera, Belén Santacruz, Daniela Piazzolla, Lieve Page-Christiaens, Diego Lerma, Ramiro Quiroga, Francisco Javier Fernández, Ignacio Herraiz.

**Methodology:** Ana Villalba, Daniela Piazzolla, Francisco Javier Fernández, Ignacio Herraiz.

**Project administration:** Ana Villalba, Daniela Piazzolla, Ignacio Herraiz.

**Resources:** Ana Villalba.

**Supervision:** Ana Villalba, M. Ángeles Sánchez-Durán, Carmen Orellana, Mercedes Sobreviela, María M. Gil, Rosa María Lobo, Irene Gómez-Manjón, Cristina Cea, Laia Pedrola Vidal, Sebastian Menao, Raquel Martin-Alonso, Gonzalo E. Quesada, Eva Albuixech, Mónica Roselló Piera, Eva Valle, Eduardo Tizzano, Alberto Galindo, Manel Mendoza, Beatriz Marcos, José Vicente Cervera, Belén Santacruz, Daniela Piazzolla, Lieve Page-Christiaens, Diego Lerma, Ramiro Quiroga, Francisco Javier Fernández, Ignacio Herraiz.

**Validation:** Ana Villalba, M. Ángeles Sánchez-Durán, Carmen Orellana, Mercedes Sobreviela, María M. Gil, Rosa María Lobo, Irene Gómez-Manjón, Cristina Cea, Laia Pedrola Vidal, Sebastian Menao, Raquel Martin-Alonso, Gonzalo E. Quesada, Eva Albuixech, Mónica Roselló Piera, Eva Valle, Eduardo Tizzano, Alberto Galindo, Manel Mendoza, Beatriz Marcos, José Vicente Cervera, Belén Santacruz, Daniela Piazzolla, Lieve Page-Christiaens, Diego Lerma, Ramiro Quiroga, Francisco Javier Fernández, Ignacio Herraiz.

**Visualization:** Ana Villalba, M. Ángeles Sánchez-Durán, Carmen Orellana, Mercedes Sobreviela, María M. Gil, Rosa María Lobo, Irene Gómez-Manjón, Cristina Cea, Laia Pedrola Vidal, Sebastian Menao, Raquel Martin-Alonso, Gonzalo E. Quesada, Eva Albuixech, Mónica Roselló Piera, Eva Valle, Eduardo Tizzano, Alberto Galindo, Manel Mendoza, Beatriz Marcos, José Vicente Cervera, Belén Santacruz, Daniela Piazzolla, Lieve Page-Christiaens, Diego Lerma, Ramiro Quiroga, Francisco Javier Fernández, Ignacio Herraiz.

**Writing – original draft:** Ana Villalba, Daniela Piazzolla, Francisco Javier Fernández, Ignacio Herraiz.

**Writing – review & editing:** Ana Villalba, Daniela Piazzolla, Francisco Javier Fernández, Ignacio Herraiz.

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
