## [Decision Letter · Decision Letter 0]

4 Jan 2026

Dear Dr. Herraiz,

Thank you for submitting your manuscript to PLOS ONE. After careful consideration, we feel that it has merit but does not fully meet PLOS ONE’s publication criteria as it currently stands. Therefore, we invite you to submit a revised version of the manuscript that addresses the points raised during the review process.

plosone@plos.org. . . . A letter that responds to each point raised by the academic editor and reviewer(s). You should upload this letter as a separate file labeled 'Response to Reviewers'.A marked-up copy of your manuscript that highlights changes made to the original version. You should upload this as a separate file labeled 'Revised Manuscript with Track Changes'.An unmarked version of your revised paper without tracked changes. You should upload this as a separate file labeled 'Manuscript'.

We look forward to receiving your revised manuscript.

Kind regards,

María Teresa Llinás

Academic Editor

PLOS One

Journal Requirements:

“This study was funded by Spanish network RD21/0012/0024: Primary care interventions to prevent maternal and child chronic diseases of perinatal and developmental origin, Instituto de Salud Carlos III, Madrid, Spain, and financed by the European Union though the Next Generation EU funds, which finance the actions of the Recovery and Resilience Facility (RRF).”

4. Thank you for stating the following in the Competing Interests/Financial Disclosure * (delete as necessary) section:

The authors report receiving reagents, technical and editorial support, through collaboration agreements between the participating centers and Illumina, Inc. Daniela Piazzolla is an employee and shareholder of Illumina Inc.

We note that you received funding from a commercial source: “Illumina, Inc”

Within this Competing Interests Statement, please confirm that this does not alter your adherence to all PLOS ONE policies on sharing data and materials by including the following statement: "This does not alter our adherence to PLOS ONE policies on sharing data and materials.” (as detailed online in our guide for authors http://journals.plos.org/plosone/s/competing-interests If there are restrictions on sharing of data and/or materials, please state these. Please note that we cannot proceed with consideration of your article until this information has been declared.

6.  For studies involving third-party data, we encourage authors to share any data specific to their analyses that they can legally distribute. PLOS recognizes, however, that authors may be using third-party data they do not have the rights to share. When third-party data cannot be publicly shared, authors must provide all information necessary for interested researchers to apply to gain access to the data. (https://journals.plos.org/plosone/s/data-availability#loc-acceptable-data-access-restrictions)   )   )   )

(1) A description of the data set and the third-party source

(2) If applicable, verification of permission to use the data set

(3) Confirmation of whether the authors received any special privileges in accessing the data that other researchers would not have

(4) All necessary contact information others would need to apply to gain access to the data

7. Please include captions for your Supporting Information files at the end of your manuscript, and update any in-text citations to match accordingly. Please see our Supporting Information guidelines for more information: http://journals.plos.org/plosone/s/supporting-information....

Reviewers' comments:

Reviewer's Responses to Questions

**Comments to the Author**

1. Does the manuscript provide a valid rationale for the proposed study, with clearly identified and justified research questions?

Reviewer #1: Partly

Reviewer #2: Yes

2. Is the protocol technically sound and planned in a manner that will lead to a meaningful outcome and allow testing the stated hypotheses?

Reviewer #1: Partly

Reviewer #2: Yes

3. Is the methodology feasible and described in sufficient detail to allow the work to be replicable?

Reviewer #1: Yes

Reviewer #2: Yes

4. Have the authors described where all data underlying the findings will be made available when the study is complete?

The PLOS Data policy requires authors to make all data underlying the findings described in their manuscript fully available without restriction, with rare exception, at the time of publication. The data should be provided as part of the manuscript or its supporting information, or deposited to a public repository. For example, in addition to summary statistics, the data points behind means, medians and variance measures should be available. If there are restrictions on publicly sharing data—e.g. participant privacy or use of data from a third party—those must be specified.requires authors to make all data underlying the findings described in their manuscript fully available without restriction, with rare exception, at the time of publication. The data should be provided as part of the manuscript or its supporting information, or deposited to a public repository. For example, in addition to summary statistics, the data points behind means, medians and variance measures should be available. If there are restrictions on publicly sharing data—e.g. participant privacy or use of data from a third party—those must be specified.

Reviewer #1: Yes

Reviewer #2: Yes

5. Is the manuscript presented in an intelligible fashion and written in standard English?

Reviewer #1: Yes

Reviewer #2: Yes

You may also provide optional suggestions and comments to authors that they might find helpful in planning their study.

Reviewer #1: The aim of the protocol study is to determine whether genome-wide non-invasive prenatal testing (GW-NIPT) provides additional clinical value compared with standard NIPT for common fetal aneuploidies in pregnancies affected by placental dysfunction.

The topic is scientifically relevant, as placental dysfunction represents a complex clinical condition in which the interpretation of cfDNA-based screening results remains challenging.

Placental mosaicism is recognized as one of the biological causes of placental dysfunction and is frequently associated with rare chromosomal abnormalities. Given that cell-free DNA analyzed by NIPT predominantly originates from trophoblastic cells, GW-NIPT theoretically holds the potential to detect confined placental mosaicism (CPM). In this context, the biological rationale of the study is sound.

It is well documented that the rate of positive NIPT screening results for rare autosomal trisomies (RATs) and large structural chromosomal abnormalities, such as copy number variants (CNVs), is higher in populations classified as high-risk for chromosomal abnormalities based on first-trimester combined screening, compared with low-risk populations. However, it is also established that the vast majority of RAT findings detected by GW-NIPT—up to approximately 50%—represent false-positive results, most likely attributable to CPM rather than true fetal involvement. This significantly limits the clinical utility of GW-NIPT, particularly in the general obstetric population, and underscores the need for careful interpretation of expanded screening results.

The proposed study is inherently limited by its restriction to a selected population already classified as being at increased risk for chromosomal abnormalities. From a clinical perspective, this significantly constrains the generalizability of the results and limits the direct impact of the findings on routine prenatal screening strategies. Consequently, the study is unlikely to substantially modify current clinical practice regarding the use of GW-NIPT in the broader obstetric population.

Nevertheless, despite this limitation, the study may still provide additional information of scientific relevance. By focusing on pregnancies characterized by placental dysfunction, the protocol has the potential to contribute to a better understanding of the biological relationship between placental mosaicism, abnormal cfDNA findings, and adverse placental phenotypes. In this sense, the value of the study lies primarily in its ability to improve the interpretation of GW-NIPT results in complex clinical scenarios, rather than in demonstrating an immediate clinical advantage over standard NIPT.

Specific comments:

The protocol aims to evaluate the clinical value of GW-NIPT compared with standard NIPT in pregnancies at increased risk of placental dysfunction, with a particular focus on the following populations:

3. Pregnancies identified as high-risk at first-trimester screening based on abnormal placental and biochemical markers, including altered PAPP-A and/or free β-hCG levels, suggestive of early placental dysfunction; and

4. Pregnancies complicated by early-onset fetal growth restriction (FGR), a condition strongly associated with impaired placentation and adverse perinatal outcomes.

The study proposes to include 321 patients in the first group and 181 patients in the second group.

Given that cell-free fetal DNA analyzed by NIPT originates predominantly from placental trophoblasts, these high-risk pregnancies represent a biologically plausible cohort in which confined placental mosaicism may contribute to abnormal cfDNA findings. However, current evidence indicates that the ability of GW-NIPT to detect CPM is highly variable and influenced by multiple biological and technical factors, including fetal fraction (FF), the level and distribution of mosaicism within the placenta, and the chromosome involved. Importantly, pregnancies complicated by early-onset FGR or abnormal first-trimester placental markers frequently exhibit reduced fetal fraction, which further limits test performance. Low fetal fraction is a well-recognized cause of NIPT test failure (“no-call” results) or reduced analytical reliability.

As a consequence, despite the expanded analytical scope of GW-NIPT, the positive predictive value for detecting CPM remains low, particularly for rare autosomal aneuploidies and subchromosomal CNVs. In this context, the sample size proposed in the protocol appears insufficient to adequately assess the clinical utility of GW-NIPT in these selected high-risk populations, especially considering that the study intends to evaluate, for each pregnancy category:

• High-risk expanded NIPT results for RATs and CNVs;

• High-risk NIPT results for common aneuploidies;

• Low-risk NIPT results.

I would recommend revising the proposed sample size. In my opinion, the number of pregnancies to be included should be increased, as the current cohort appears insufficient to draw robust and clinically meaningful conclusions. A minimum of 500 and 600 patients should be included in the first and second study groups, respectively.

Furthermore, the protocol would benefit from the inclusion of the following clarifications:

4. A clear specification of the gestational age at which NIPT sampling is performed. Gestational timing critically affects fetal fraction and analytical reliability, particularly in pregnancies characterized by placental insufficiency, early-onset FGR, or abnormal first-trimester screening markers.

In addition, the protocol does not specify:

• how no-call or failed NIPT results will be managed,

• whether repeat blood sampling is planned following an initial test failure,

• nor how repeated GW-NIPT results will be handled in the final analysis.

5. A justification for the requirement of 20 mL of maternal blood instead of the standard 10 mL volume.

In conclusion, the proposed study is scientifically sound, and both the objectives and inclusion criteria are appropriate. However, expansion of the study cohort and clarification of the points outlined above are necessary. With these modifications, the protocol would be suitable for publication.

Reviewer #2: The proposed protocol is feasible and has high potential to generate clinically relevant results due to its methodological design, the relevance of the study population, and the use of advanced technologies. It is a prospective, multicenter, observational study, which allows real-time data collection. The participation of six hospitals in different regions of Spain increases the representativeness of the sample and the external validity of the findings.

The use of expanded NIPT based on whole-genome sequencing (VeriSeq™ NIPT Solution v2) is an innovative and clinically relevant element, as it enables the detection not only of common trisomies but also of rare autosomal trisomies and copy number variations (CNVs) greater than 7 Mb. This technology offers advantages over traditional invasive methods, such as amniocentesis, especially in cases where these are not feasible or are declined by the patient. The protocol includes a sample size calculations based on previous estimates of chromosomal abnormality and CPM prevalence in similar populations, ensuring 80% statistical power to detect significant differences. A

Finally, the study complies with ethical and regulatory standards, including approval by an ethics committee, informed consent, and data protection measures in accordance with the General Data Protection Regulation (GDPR). It also includes postnatal follow-up for six months, adding value to the clinical interpretation of the findings.

Taken together, these elements confirm that the protocol is feasible, and capable of providing relevant evidence to improve the interpretation and clinical use of expanded NIPT in pregnancies with suspected placental dysfunction.

.

Reviewer #1: No

Reviewer #2: **Yes:** Isabel Hernandez GarcíaIsabel Hernandez GarcíaIsabel Hernandez GarcíaIsabel Hernandez García

---

## [Author Response · Author response to Decision Letter 1]

6 Feb 2026

PONE-D-25-50467 — Response to Reviewers

Manuscript: Protocol for a prospective observational cohort study to assess clinical applications of expanded noninvasive prenatal testing (NIPT) in pregnancies with placental dysfunction

Journal: PLOS ONE, Manuscript ID: PONE-D-25-50467

To the editor and reviewers,

Thank you for your thoughtful review of our manuscript. With this letter, we provide a response to each of the points raised and have made the necessary changes to the cover letter and manuscript files, which are both uploaded with this letter. To facilitate your review of our responses, the journal’s and reviewers’ comments are included below and highlighted in blue.

Additionally, we made minor edits to lines 323 and 327-328 of the revised manuscript for clarity.

We look forward to hearing your final decision on this protocol manuscript.

Ana Villalba

Journal Requirements

1) Formatting & file naming

Response: Thank you for your guidance on journal requirements. We have reviewed the documents cited above and amended the manuscript and file names in accordance with these requirements. In doing so, we noticed a numerical error in the annotation of the author affiliations and have corrected this in track changes mode.

2) Role of funders

Thank you for stating the following financial disclosure:

“This study was funded by Spanish network RD21/0012/0024: Primary care interventions to prevent maternal and child chronic diseases of perinatal and developmental origin, Instituto de Salud Carlos III, Madrid, Spain, and financed by the European Union though the Next Generation EU funds, which finance the actions of the Recovery and Resilience Facility (RRF).”

Response: We have updated the financial disclosures in the manuscript text as requested (lines 50-60) and included the role of funders statement in the cover letter.

3) Funding information

We note that the grant information you provided in the ‘Funding Information’ and ‘Financial Disclosure’ sections do not match.

Response: We have amended the funding and competing interests’ sections of the manuscript as requested (lines 50-68).

4) Competing interests

Thank you for stating the following in the Competing Interests/Financial Disclosure * (delete as necessary) section:

The authors report receiving reagents, technical and editorial support, through collaboration agreements between the participating centers and Illumina, Inc. Daniela Piazzolla is an employee and shareholder of Illumina Inc.

We note that you received funding from a commercial source: “Illumina, Inc”

Response: We have amended the competing interests section of the manuscript text as requested (lines 62-68) and included this information in the cover letter.

5) Ethics statement

Your ethics statement should only appear in the Methods section of your manuscript. If your ethics statement is written in any section besides the Methods, please move it to the Methods section and delete it from any other section. Please ensure that your ethics statement is included in your manuscript, as the ethics statement entered into the online submission form will not be published alongside your manuscript.

Response: The ethics statement is provided at the end of the materials and methods section of the manuscript (lines 389-392).

6) Data Availability

For studies involving third-party data, we encourage authors to share any data specific to their analyses that they can legally distribute. PLOS recognizes, however, that authors may be using third-party data they do not have the rights to share. When third-party data cannot be publicly shared, authors must provide all information necessary for interested researchers to apply to gain access to the data. (https://journals.plos.org/plosone/s/data-availability#loc-acceptable-data-access-restrictions)

(1) A description of the data set and the third-party source

(2) If applicable, verification of permission to use the data set

(3) Confirmation of whether the authors received any special privileges in accessing the data that other researchers would not have

(4) All necessary contact information others would need to apply to gain access to the data

Response: The data availability statement in the manuscript file has been modified as requested to more fully describe the data sharing plan and the conditions under which data that cannot be publicly shared may be available (lines 70-78).

7) Supporting Information captions

Please include captions for your Supporting Information files at the end of your manuscript, and update any in-text citations to match accordingly. Please see our Supporting Information guidelines for more information: http://journals.plos.org/plosone/s/supporting-information.

Response: A caption for the sole supporting information file for this manuscript has been added to the supporting information section at the end of the manuscript (lines 411-412). The in-text citation of the supporting information has been updated as prescribed (lines 283-284).

8) Citation recommendations

Response: No additional citations were recommended by the reviewers or editor.

Comments to the Author

1. Does the manuscript provide a valid rationale for the proposed study, with clearly identified and justified research questions?

Reviewer #1: Partly

Reviewer #2: Yes

Response: Thank you for your thoughtful reviews of our manuscript. We appreciate your insights as to the relevance of this work to the field.

2. Is the protocol technically sound and planned in a manner that will lead to a meaningful outcome and allow testing the stated hypotheses?

Reviewer #1: Partly

Reviewer #2: Yes

Response: We endeavoured to describe the methods in as much detail as possible to allow fair assessment of the protocol.

3. Is the methodology feasible and described in sufficient detail to allow the work to be replicable?

Reviewer #1: Yes

Reviewer #2: Yes

Response: We appreciate your assessment that our study protocol is feasible and described in sufficient detail to allow replication by others.

4. Have the authors described where all data underlying the findings will be made available when the study is complete?

Reviewer #1: Yes

Reviewer #2: Yes

Response: We appreciate your review of the data sharing plan. At the request of the journal, we have provided additional details (lines 70-78).

5. Is the manuscript presented in an intelligible fashion and written in standard English? PLOS ONE does not copyedit accepted manuscripts, so the language in submitted articles must be clear, correct, and unambiguous. Any typographical or grammatical errors should be corrected at revision, so please note any specific errors here.

Reviewer #1: Yes

Reviewer #2: Yes

Response: Thank you for your assessment.

6. Review Comments to the Author

Reviewer #1: The aim of the protocol study is to determine whether genome-wide non-invasive prenatal testing (GW-NIPT) provides additional clinical value compared with standard NIPT for common fetal aneuploidies in pregnancies affected by placental dysfunction.

The topic is scientifically relevant, as placental dysfunction represents a complex clinical condition in which the interpretation of cfDNA-based screening results remains challenging. Placental mosaicism is recognized as one of the biological causes of placental dysfunction and is frequently associated with rare chromosomal abnormalities. Given that cell-free DNA analyzed by NIPT predominantly originates from trophoblastic cells, GW-NIPT theoretically holds the potential to detect confined placental mosaicism (CPM). In this context, the biological rationale of the study is sound.

It is well documented that the rate of positive NIPT screening results for rare autosomal trisomies (RATs) and large structural chromosomal abnormalities, such as copy number variants (CNVs), is higher in populations classified as high-risk for chromosomal abnormalities based on first-trimester combined screening, compared with low-risk populations. However, it is also established that the vast majority of RAT findings detected by GW-NIPT—up to approximately 50%—represent false-positive results, most likely attributable to CPM rather than true fetal involvement. This significantly limits the clinical utility of GW-NIPT, particularly in the general obstetric population, and underscores the need for careful interpretation of expanded screening results.

The proposed study is inherently limited by its restriction to a selected population already classified as being at increased risk for chromosomal abnormalities. From a clinical perspective, this significantly constrains the generalizability of the results and limits the direct impact of the findings on routine prenatal screening strategies. Consequently, the study is unlikely to substantially modify current clinical practice regarding the use of GW-NIPT in the broader obstetric population.

Nevertheless, despite this limitation, the study may still provide additional information of scientific relevance. By focusing on pregnancies characterized by placental dysfunction, the protocol has the potential to contribute to a better understanding of the biological relationship between placental mosaicism, abnormal cfDNA findings, and adverse placental phenotypes. In this sense, the value of the study lies primarily in its ability to improve the interpretation of GW-NIPT results in complex clinical scenarios, rather than in demonstrating an immediate clinical advantage over standard NIPT.

Specific comments:

The protocol aims to evaluate the clinical value of GW-NIPT compared with standard NIPT in pregnancies at increased risk of placental dysfunction, with a particular focus on the following populations:

• Pregnancies identified as high-risk at first-trimester screening based on abnormal placental and biochemical markers, including altered PAPP-A and/or free β-hCG levels, suggestive of early placental dysfunction; and

• Pregnancies complicated by early-onset fetal growth restriction (FGR), a condition strongly associated with impaired placentation and adverse perinatal outcomes.

The study proposes to include 321 patients in the first group and 181 patients in the second group. Given that cell-free fetal DNA analyzed by NIPT originates predominantly from placental trophoblasts, these high-risk pregnancies represent a biologically plausible cohort in which confined placental mosaicism may contribute to abnormal cfDNA findings. However, current evidence indicates that the ability of GW-NIPT to detect CPM is highly variable and influenced by multiple biological and technical factors, including fetal fraction (FF), the level and distribution of mosaicism within the placenta, and the chromosome involved. Importantly, pregnancies complicated by early-onset FGR or abnormal first-trimester placental markers frequently exhibit reduced fetal fraction, which further limits test performance. Low fetal fraction is a well-recognized cause of NIPT test failure (“no-call” results) or reduced analytical reliability. As a consequence, despite the expanded analytical scope of GW-NIPT, the positive predictive value for detecting CPM remains low, particularly for rare autosomal aneuploidies and subchromosomal CNVs. In this context, the sample size proposed in the protocol appears insufficient to adequately assess the clinical utility of GW-NIPT in these selected high-risk populations, especially considering that the study intends to evaluate, for each pregnancy category:

• High-risk expanded NIPT results for RATs and CNVs;

• High-risk NIPT results for common aneuploidies;

• Low-risk NIPT results.

I would recommend revising the proposed sample size. In my opinion, the number of pregnancies to be included should be increased, as the current cohort appears insufficient to draw robust and clinically meaningful conclusions. A minimum of 500 and 600 patients should be included in the first and second study groups, respectively.

Response: Thank you for raising this important point. We acknowledge the concerns regarding the rarity of RATs and subchromosomal CNVs and the potential impact on precision. For early onset FGR (eoFGR), prior literature [14] reports a 9%–16% risk of CPM-related chromosomal abnormalit

---

## [Decision Letter · Decision Letter 1]

25 Feb 2026

Protocol for a prospective observational cohort study to assess clinical applications of expanded noninvasive prenatal testing (NIPT) in pregnancies with placental dysfunction

PONE-D-25-50467R1

Dear Dr. Herraiz,

We’re pleased to inform you that your manuscript has been judged scientifically suitable for publication and will be formally accepted for publication once it meets all outstanding technical requirements.

Kind regards,

María Teresa Llinás

Academic Editor

PLOS One

Additional Editor Comments (optional):

Reviewers' comments:

Reviewer's Responses to Questions

**Comments to the Author**

1. Does the manuscript provide a valid rationale for the proposed study, with clearly identified and justified research questions?

Reviewer #1: Yes

2. Is the protocol technically sound and planned in a manner that will lead to a meaningful outcome and allow testing the stated hypotheses?

Reviewer #1: Yes

3. Is the methodology feasible and described in sufficient detail to allow the work to be replicable?

Reviewer #1: Yes

4. Have the authors described where all data underlying the findings will be made available when the study is complete?

The PLOS Data policy requires authors to make all data underlying the findings described in their manuscript fully available without restriction, with rare exception, at the time of publication. The data should be provided as part of the manuscript or its supporting information, or deposited to a public repository. For example, in addition to summary statistics, the data points behind means, medians and variance measures should be available. If there are restrictions on publicly sharing data—e.g. participant privacy or use of data from a third party—those must be specified.requires authors to make all data underlying the findings described in their manuscript fully available without restriction, with rare exception, at the time of publication. The data should be provided as part of the manuscript or its supporting information, or deposited to a public repository. For example, in addition to summary statistics, the data points behind means, medians and variance measures should be available. If there are restrictions on publicly sharing data—e.g. participant privacy or use of data from a third party—those must be specified.

Reviewer #1: Yes

5. Is the manuscript presented in an intelligible fashion and written in standard English?

Reviewer #1: Yes

You may also provide optional suggestions and comments to authors that they might find helpful in planning their study.

Reviewer #1: The authors have addressed all the comments and observations in a precise and thorough manner, providing the necessary clarifications and revisions.

In light of the detailed responses and the improvements made, I believe the manuscript now meets the standards required for publication.

.

Reviewer #1: **Yes:** Francesca SpinellaFrancesca SpinellaFrancesca SpinellaFrancesca Spinella

---

## [Editor Report · Acceptance letter]

PONE-D-25-50467R1

PLOS One

Dear Dr. Herraiz,

I'm pleased to inform you that your manuscript has been deemed suitable for publication in PLOS One. Congratulations! Your manuscript is now being handed over to our production team.

Kind regards,

on behalf of

Dr. María Teresa Llinás

Academic Editor

PLOS One